# Geotourism in the Cilento, Vallo di Diano and Alburni UNESCO Global Geopark (Southern Italy): The Middle Bussento Karst System

**Ettore Valente** [1],[*] , **Antonio Santo** [2] , **Domenico Guida** [3] **and Nicoletta Santangelo** [1]

1   Department of Earth, Environmental and Resource Science, DISTAR, University of Naples Federico II, via Cinthia 21, 80126 Naples, Italy; nicsanta@unina.it
2   Dipartimento di Ingegneria Civile, Edile e Ambientale, DICEA, University of Naples Federico II, piazzale Tecchio 80, 80125 Naples, Italy; antonio.santo@unina.it
3   Department of Civil Engineering and Consorzio inter-Universitario per la previsione e prevenzione dei Grandi Rischi (CUGRI), University of Salerno, Campus of Fisciano, Via Giovanni Paolo II, 132–84084 Fisciano, Italy; dguida@unisa.it
*   Correspondence: ettore.valente@unina.it

**Abstract:** In this paper we want to stress the role of geotourism as a means to promote environmental education and, on occasion, as a way to increase the touristic interest of an area. Geoparks are certainly the territory where geotourism can be best exploited. We propose a geoitinerary to discover the amazing, but poorly known, Middle Bussento Karst System, with the blind valley of the Bussento River, in the southeast of the Cilento, Vallo di Diano and Alburni United Nations Educational, Scientific and Cultural Organization (UNESCO) Global Geopark. This is the only example, in Southern Italy, of a stream sinking underground and it is the second longest subsurface river path in Italy, making this a core area of the Geopark. We combined field surveys and literature data to create a geoitinerary that can be useful in helping to promote this site. This geoitinerary is applicable to both simple generic visitors and geo-tourists and has an educational purpose, especially in explaining the significance and the fragility of karst areas in terms of environmental protection. Moreover, it may represent a sort of stimulus for the growth of touristic activity in this inner area of the Geopark.

**Keywords:** geotourism; geomorphosites; environmental education; Cilento, Vallo di Diano and Alburni Geopark; Middle Bussento Karst System

---

## 1. Introduction

### 1.1. Geoparks and Geotourism: An Overview

In recent decades, the growing awareness to protect nature from human impact has led to the diffusion of several legislative and social actions, both at international and national levels. The protection of nature is, for example, clearly expressed in article no. 9 of the Italian Constitution [1], from December 1947. In Italy, one of the first laws to address the protection of nature is the Galasso Law (law no. 431, 22 August 1985). This law introduced the concept of "protected areas" in Italian territory, namely seas, rivers, mountain areas from 1200 m above sea level (m a.s.l.), volcanoes, forests, glaciers, national parks, and archaeological areas.

At the international level, in 1972, the United Nations Educational, Scientific and Cultural Organization (UNESCO) adopted the convention "Concerning the Protection of the World Cultural and Natural Heritage". This convention identified 1500 sites all over the world to be included in the World Heritage List, due to outstanding values of their natural and cultural features ([2,3] and references

therein). Currently, the sites included in the World Heritage List are mainly cultural sites (869 of the 1121 sites), whereas a lower number of natural sites with a high geological and geomorphological scientific and scenic value are included in the list (213 of the 1121 sites; [4]). To overpass this low number of natural sites included in the World Heritage List and considering the large number of sites in the world with high geo-scientific significance, in 1997 the Division of Earth Science at UNESCO proposed a new program called the "UNESCO's Geoparks Programme" [5,6]. Prior to the creation of this program, the idea of establishing a Geoparks network was settled for the first time during the 30th International Geological Congress in Beijing in 1996 [6]. A Geopark is defined as "a territory with well-defined limits that has a large enough surface area for it to serve local economic development. The Geopark comprises a number of geological-paleontological heritage sites of special scientific importance, rarity or beauty; it may not be solely of geological-paleontological significance but also of archaeological, ecological, historical or cultural value" [2]. The first 25 Geoparks were established in Europe and China, and in 2004 they formed the UNESCO Global Geoparks Network [2,6]. It is worth mentioning that the first protected area in the world was established at Yellowstone National Park in North America in 1872 [7]. Currently, only three Canadian national parks have gained the title of 'Geoparks' in North America [8]. UNESCO Global Geoparks are defined as "single, unified geographical areas where sites and landscapes of international geological significance are managed with a holistic concept of protection, education and sustainable development" [9]. UNESCO Global Geoparks focus their activities on raising awareness in the local community about the geological heritage of the area, promoting the concept that the landscape is a dynamic element, stimulating sustainable Geotourism, and encouraging the protection of geological resources [9].

　　The definition of Geopark includes a clear reference to local economic development, so geological and geomorphological features within a Geopark must be considered as crucial geological resources ([10] and references therein) forming the so-called geoheritage [11–13]. Growth of the local economy must be practiced through sustainable management strategies that seek to develop geotourism by attracting an increasing number of visitors. The concept of geotourism has widely diffused in the last decades [14–19] among international papers covering many countries throughout the world ([20] and references therein). Moreover, Dowling and Newsome [21] highlighted that geotourism can be defined from both a geological and a geographical point of view. The geological definition of geotourism was first proposed by Hose [14,22] who defined it as "the provision of interpretive and service facilities to enable tourists to acquire knowledge and understanding of the geology and geomorphology of a site beyond the level of a mere aesthetic appreciation". This definition has since been refined by the same author [23,24]. In 2006 another definition was proposed by Dowling and Newsome [16] who introduced the concept of scale, suggesting that geotourism focuses on both small geological and paleontological sites and large landforms and landscapes. Newsome and Dowling [17] pointed out that geotourism is a form of tourism focused on geology and landscape that can be carried out either by independent visits or guided tours. Hose [15] suggested that geotourism is underpinned by the so-called 3G's, namely geoconservation, geohistory, and geo-interpretation. In contrast, in 2000, the National Geographic Society of the United States of America defined geotourism as "tourism that sustains or enhances the geographical character of a place-its environment, culture, aesthetics, heritage, and the well-being of its residents" ([21] and references therein). In 2006 a new definition was set by Pralong [25] that highlighted the emotional aspects of geotourism and introduced the concept of geomarketing, thus placing geotourism as a component of the regional economy. Subsequently, in 2011, during the International Congress on Geotourism at Arouca (Portugal) a new definition was proposed that included the term "geology" in the geographical definition of geotourism [26]. These two points of view on what is geotourism suggest considering it either as a "type or form" of tourism (geological definition) or as an "approach" to tourism (geographical definition) [21]. In this paper we adopt the geological definition of geotourism.

　　To discuss geoheritage and geotourism it is necessary to recognize geosites and geomorphosites. The former are "portions of the geosphere that present a particular importance for the comprehension

and reconstruction of the history of the Earth, climate and life" [27]. The latter are "geomorphological landforms that have acquired a scientific, cultural/historical, aesthetic and/or social/economic value due to human perception or exploitation" [28]. Moreover, the assessment methods of geomorphosites and their importance in geotourism and geoheritage have been widely discussed in international papers [29–45].

The concept of geotourism has been widely diffused in Italy. With its ten UNESCO Geoparks, is the third country in the world with the highest number of Geoparks, being preceded only by China and Spain [8]. Addressing Geotourism and Earth Science education in Italy, numerous scientific papers have been produced which deal with geosites and geomorphosites inventory in many areas of the national territory [46–53]. Most of these papers focus on Central and Northern Italy, and less international papers have been published which deal with geosites and geotourism in Southern Italy [54–58]. In particular, few international papers [59] and conference proceedings [60] have addressed Cilento, Vallo di Diano and Alburni Geopark and only some national papers [61–63] deal with geosites inventory and geotourism in the territory of the Geopark.

### 1.2. The Cilento, Vallo di Diano and Alburni UNESCO Global Geopark

The Cilento, Vallo di Diano and Alburni Geopark (Figure 1) is among the largest Italian Geoparks, covering an area of 181,048 ha that includes 80 municipalities in the province of Salerno, with a population of ~280,000 inhabitants. The Geopark was firstly established as a national park, namely the National Park of Cilento and Vallo di Diano, in 1991 under the law 394/91. In 1997 the national park was included in the Man and the Biosphere Program of UNESCO and it became part of UNESCO's World Heritage List in 1998. Then in 2010 during the 9th European Conference of Geoparks in Lesvos (Greece), it gained the title of Geopark and became part of the European Geopark Network. The Geopark finally gained the title of UNESCO Global Geopark in 2015.

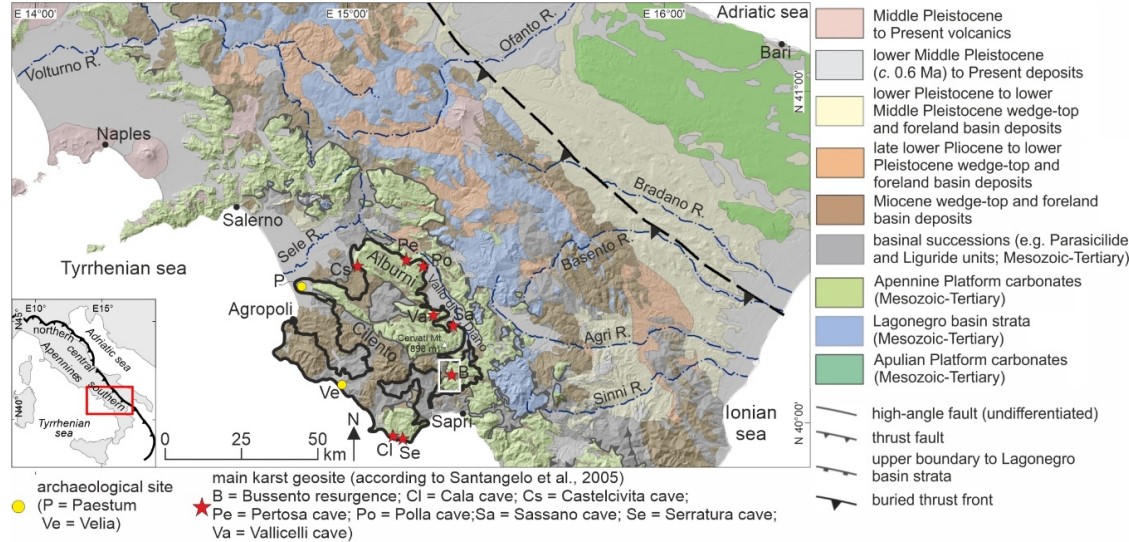

**Figure 1.** Geological map of the Southern Apennines (modified from [64]). Thick black line indicates the boundary of the Cilento, Vallo di Diano and Alburni Geopark (from [65]). White box in the southest (SE) corner of the Geopark indicates the location of Figure 2.

The Geopark of Cilento, Vallo di Diano and Alburni is located in the southern part of the Campania region. It extends from the coastal areas between the towns of Agropoli, to the north, and Sapri, to the south, up to the mountain ridges of the Southern Apennines chain, towards the east, reaching a maximum elevation of 1898 m above sea level (m a.s.l.) at the peak of Mt. Cervati (Figure 1). The Geopark is characterized by a complex geological setting, with large outcrops of permeable carbonate units in the inner mountainous landscape and less permeable or even impermeable wedge-top and

basinal units towards the coast (Figure 1). Its complex geological setting results in a large variety of stratigraphical and geomorphological features, whose scientific importance and beautiful scenery are undisputed [59]. While the coastal area with its beautiful beaches and the well-known archeological sites of Paestum and Velia (Figure 1) is a continuously growing touristic attraction, with thousands of visitors per year, the inner portion of the park is less visited and there are few initiatives promoted to invert this trend.

Inventory, evaluation, and protection of the wide geological estate of the Geopark have been carried out by the competent authorities and are listed in the official catalogue promoted by national [66] and regional institutions [67], as well as by the Geopark itself. Moreover, Santangelo et al. [59] identified a total of 263 sites of geological interest, 32% of which are geomorphosites.

In this paper, we investigate a poorly known portion of the Geopark that is placed along its southeastern border. This portion of the Geopark is mainly made up of Mesozoic carbonate platform successions (Figure 1) which are strongly karstified and host some of the most important water reservoirs of the region with springs that have a total discharge of more than 20 m$^3$/s [68,69]. Karst processes are so well represented in the area both at the surface (doline, polje, gorges, ponors, karst springs) and underground (horizontal and vertical cave/karst systems [70]). Many of these karst landforms are listed in the official catalogue of the Geopark [71] as geomorphosites according to different perspectives: some cave, for instance, preserve important archaeological records [72], while others represent the longest or the deepest karst systems of the area; meanwhile others are exemplary or have a particular didactic value [59].

Unfortunately, up until now, few initiatives have been carried out to promote this significant geological estate. The foundation of the Musei Integrati dell'Ambiente (MIDA) museum [73], which is part of the touristic Pertosa cave management system (see Figure 1 for location), has, among others, a sector dedicated to the explanation of karst processes. At the same time, the proposition of geotouristic itineraries in this portion of the Geopark is still at an early stage with only some papers addressed to both Italian (itineraries n. 11, 12, and 13 in [61–63]) and international tourists [54]. Moreover, Aloia and colleagues [61] briefly discuss the importance of correct management strategies of the Geopark as a tool to direct touristic flows from the coastal area to the inner, hilly and mountainous areas.

Our study aims to focus attention on the karst geomorphosite of the middle Bussento river system. This site is the only example of stream sinking underground in Southern Italy [59] and is the second longest underground river in Italy, being preceded only by the Timavo River in North-East Italy [74]. This geomorphosite includes a system of ponors, the largest of which is the La Rupe ponor, where the Bussento River sinks, and the Bussento Resurgence, where the Bussento River reemerges after a ~4 km long subsurface path. The area is already the object of some, local scale, touristic promotion activities such as the foundation of the Museo Virtuale (MU.VI.), a virtual museum conceived as an educational and scientific center, managed by the Caselle in Pittari administration, where teaching materials are organized for presentation by means of visual technologies (multi-touch screen, 3D room for virtual reconstruction). Yet despite the high scientific and educational values of this portion of the Bussento river valley, an adequate, comprehensive geotourism policy has not been assessed by the local administration.

In this context, we attempt to contribute to an increase in the knowledge of this fascinating portion of the Geopark by the promotion of a comprehensive geoitinerary, which should serve both as a scientific and educational instrument at inter-municipality scale. The main aims of this work are: (i) contributing to explain how karst processes can make a river disappear underground; (ii) discussing the importance of karst aquifer and the main environmental implications connected to the communication between surface and underground waters; (iii) increasing curiosity about this site, helping to promote the integrate management of this inner portion of the Geopark as a touristic attraction and thus helping to grow the local and the district economy.

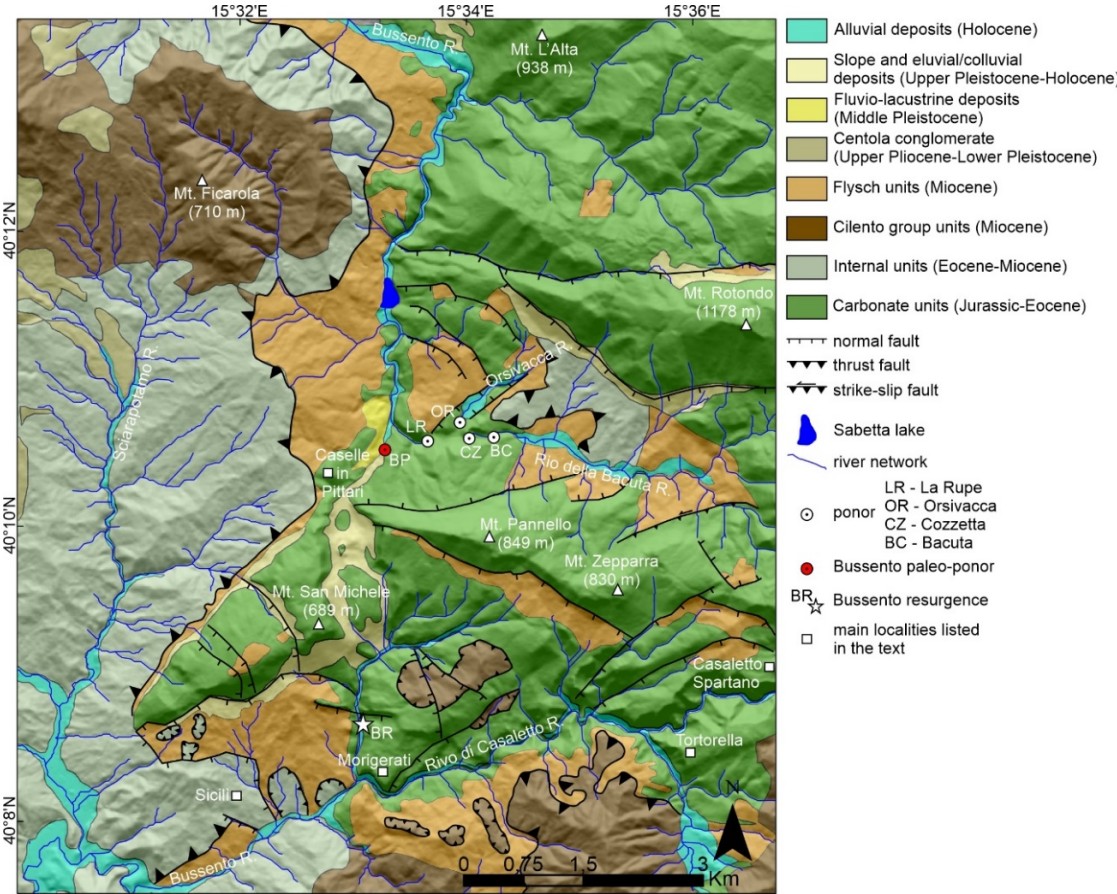

**Figure 2.** Geological map of the Middle Bussento Karst System (modified from [75]).

## 2. Study Area: The Middle Bussento Karst System

The study area is characterized by the occurrence, in a very restricted area, of Jurassic to Eocene carbonate units, of Eocene to Miocene internal units, and of Miocene flysch and Cilento units (Figure 2). Carbonate units consist of inner platform limestone with high fossils content (rudists and gasteropods) passing upward to open shelf limestone with interlayered marls and clays. Internal units consist of clays with interbedded quartz-rich and feldspar-rich sandstones passing upwards through calcilutite, calcarenite, and calcirudite with cherty lists and nodules and with interbedded quartz-arenite and marls strata. Miocene flysch units consist of a fining-upward sequence of thrust-top deposits whereas the Miocene Cilento group consists of a coarsening upward sequence of a wedge-top basin [75].

The actual structural setting of the area derives from its complex tectonic evolution with internal units thrusted over the carbonate units that were covered by the flysch units. The tectonic setting is completed by the occurrence of E-W, NW-SE, and NW-SE normal faults that articulate the topography. Moreover, flysch units have been eroded from the high carbonate peaks and are preserved only along the main valleys [75]. The complex tectonic evolution of the area is discussed in other studies [76–82].

The main river flowing in the study area is the Bussento River (Figure 2). This river originates from Mt. Cervati and Mt. Ficarola, following a mainly NW-SE orientation in the upper portion of its basin where it carves impermeable units. At the base of Mt. L'Alta (Figure 2) the Bussento River flows along a N-S trend and starts carving into the Mesozoic carbonate units. Then, it suddenly disappears near the village of Caselle in Pittari in the so called La Rupe ponor to flow out again ~4 km to the south, in the so-called Bussento Resurgence, near the village of Morigerati (Figure 2). To the east of the middle Bussento River there are two more rivers—Orsivacca River and Rio della Bacuta River—that, after flowing in the impermeable flysch and internal units, suddenly disappear underground when

they carve the carbonate units. Moreover, the Orsivacca River ends in the Orsivacca ponor whereas the Rio della Bacuta River has both a fossil and an active ponor, respectively named the Cozzetta and Bacuta ponors (Figure 2). D'Elia et al. [83] suggested that the Orsivacca and the Rio della Bacuta rivers were left tributaries of the Bussento River and that they were separated from it following the ponor regression mechanism [84,85]. The separation of the Orsivacca and the Rio della Bacuta rivers from the Bussento River is also testified by a paleo-valley placed in-between the relative ponors. When these three river segments were connected, the Bussento River should have had a higher discharge that could justify the presence of a large fossil ponor, named the Bussento paleo-ponor (Figure 2), which is placed to the southwest of the active La Rupe ponor [83].

It is worth noting that the Bussento River is intercepted by the Sabetta reservoir, an artificial basin created in 1958, which serves a hydroelectrical powerplant. Therefore, the Bussento River discharge downstream of the Sabetta reservoir depends on the amount of water released for operational activities and the amount of water coming from the residual drainage basin between the Sabetta Lake and the La Rupe ponor [86].

Regarding the hydrogeological setting of the area, the main aquifer is part of the carbonate structure of the Salice-Coccovello Mts, which drains towards the basal spring systems of Morigerati, located at about 90 m a.s.l., which has a total discharge of 1.5 $m^3$/s. Hydrogeological analysis has proved a subsurface connection between the La Rupe, Orsivacca, Cozzetta, and Bacuta ponors with the Morigerati springs, that are all part of the so-called Middle Bussento Karst System [87].

Recent scientific interdisciplinary research has studied the Middle Bussento Karst System as one of the most interesting karst systems in Southern Italy, spanning from the singular karst back-flooding to the karst pulse floods, suggesting it as experimental UNESCO karst basin [86] for geodiversity conservation, protection, and promotion.

## 3. Materials and Methods

As we already discussed in Section 1, Geoparks are intended to not only protect the geoheritage of an area but also to promote a holistic concept of education and sustainable development. Within a Geopark, geotourism represents the most useful tool to enable tourists to acquire knowledge and understanding of the geology and geomorphology of a site. With these concepts in mind, we focused our attention on the inner area of the Cilento, Alburni and Vallo di Diano Geopark, and chose one of the geosites already listed in the official catalogue by the competent authorities. We used this site as an example of how geotourism may be a useful tool to increase the touristic attraction of an area and promote environmental education.

The inner part of the Geopark is mainly made of carbonate rocks and for this reason our choice fell among karstic geomorphosites. We selected the Middle Bussento Karst System because of its uniqueness as the sole example of an underground river in Southern Italy. Moreover, it also has a high didactic value because it is representative of the extreme significance and sensitivity of the karst environment as water reservoirs.

We collected and revised all literature data about the geological, geomorphological, hydrogeological, and speleological setting of the Middle Bussento Karst System and planned the geoitinerary.

Field surveys (5 days in total) were carried out to detail the technical issues of the geoitinerary. It is 17 km long and can be done entirely by walking or it is possible to move by car from one stop to the next. For those who prefer the latter option, we determined the path length for each stop. It indicates the distance from the place where tourists can leave the car until they reach the ponor. In addition, the duration (in hours) of both the entire geoitinerary and every single path is indicated, together with the differences in elevation of every path and the main geological and geomorphological features that can be admired along each path.

To emphasize the scientific and educational importance of the proposed itinerary we prepared some sketches to provide tourists with a complete overview of both the surface and subsurface setting

of the Middle Bussento Karst System. These sketches include a 10 m digital terrain model (DTM) of the investigated area and a 3D view of the topography (both from the south and from the east). The 10 m DTM was created from elevation data (both contour lines and elevation points) derived from a detailed scale topographic map (Technical Map of the Campania Region, at scale 1:5000). Elevation data were imported in a Geographic Information System (GIS) software (ArcGis 10.7©, Redlands, CA, USA) and interpolated by means of the Topo to Raster tool to obtain the DTM that, successively, was used to derive the hillshade map (by means of the 3D Analyst tool in ArcGis) and the river network (by means of the Hydrology tool in ArcGis). Both the hillshade map and the river network were then imported in the ArcScene module of ArcGis to obtain two 3D views of the topography that focused on the area between the Sabetta Lake, to the north, and the Bussento Resurgence, to the south. These two 3D views, respectively from the south and from the west, were then imported in Corel Draw© (Ottawa, MI, Canada) to produce the final sketch.

To analyze the potential of the proposed geoitinerary, we carried out a classical SWOT (strengths, weaknesses, opportunities and threats) analysis. Finally, to assess the geotouristic value of the Middle Bussento Karst System geomorphosite, we calculated the index proposed by Pica et al. [88] named the value of a site for geotourism (VSG). It results from the following equation:

$$VSG = RP + RR + SCE + SAC + AC \qquad (1)$$

RP is the representativeness index, RR is the rarity index, SCE is the scenic-aesthetic value, SAC is the historical-archaeological-cultural value, and AC is the accessibility index. Each index has a maximum score of 5 so that the highest value of the VSG is 25. The scores derive from the geosites' characteristics reported in Table 1. According to Pica et al. [88], VSG values lower than 8 indicate that the site has low touristic potential, VSG values between 9 and 16 indicate a medium touristic potential, and values between 17 and 25 indicate a high touristic potential.

**Table 1.** Scores of the indexes used for the evaluation of the value of a site for geotourism (VSG) index (from Pica et al. [88]). RP = representativeness index; RR = rarity index; SCE = scenic-aesthetic value; SAC = historical-archaeological-cultural value; AC = accessibility index.

| Value of a Site for Geotourism | | |
|:---:|:---:|:---:|
| **VSG = RP + RR + SCE + SAC + AC, VSG max = 25** | | |
| **Attributes** | **Values** | |
| **Representativeness RP** | **0, 1, 3, 5** | |
| Ideal model correspondence | 5, 3, 3, 1, 0 | |
| Peculiarity (lithostratigraphy, carsism, hydrology, paleontology, geomorphology, structural geology, mineralogy) | 5, 3, 3, 1, 0 | } |
| Typicality | 5, 3, 3, 1, 0 | |
| Interest peculiarity | 5, 3, 3, 1, 0 | |
| **Rareness RR** | **0, 1, 3, 5** | |
| Geographical range | local, regional, national, international | Two-way table |
| Frequency | 5, 4, 3, 1, 0 | |

Table 1. *Cont.*

| Value of a Site for Geotourism | |
| --- | --- |
| **VSG = RP + RR + SCE + SAC + AC, VSG max = 25** | |
| **Attributes** | **Values** |
| **Scenic-aesthetic SCE** | **0, 1, 3, 5** |
| Viewpoints | 5, 3, 1, 0 |
| Cromatic contrast | 5, 3, 1, 0 |
| Landform queerness | 5, 0 — summarize intervals |
| **Historical-archeological-cultural SAC** | **0, 1, 3, 5** |
| National restriction | 3, 5 (area, geosite) |
| Regional/local restriction | 1, 3 (area, geosite) |
| Protected area | 3, 5, 1 |
| Other (archeological, monumental, architectural values; legends, stories, tradition; toponym) | 2, 2, 1 — summarize intervals |
| **Accessibility AC** | **0, 1, 3, 5** |
| Ways to approach the site | 5, 3, 1 |
| Difficulty to approach the site | 5, 4, 3, 1 |
| Services | 5, 4, 3, 1, 0 — summarize intervals |

## 4. Results

### 4.1. Karst Landforms (Ponors, Blind Valleys, Resurgences, and Karst Springs): Importance for Environmental Education

Karst processes occur on Earth's surface in any place where soluble rocks like limestones or gypsum crop out and are exposed to the action of meteoric waters. The latter have a natural content of $CO_2$ that increases during percolation in the soil. For this reason, they can dissolve soluble rocks, creating spectacular morphologies both at the surface and underground [89] (Figure 3). Dissolution makes the rocks highly permeable, allowing for the circulation and accumulation of water underground. A mountain made up of permeable and soluble rock, like limestone, behaves as a sponge, adsorbing all the meteoric waters dropping above its surface. For this reason, at depth, a subterranean water body originates, called by geologists as a "water table". The mountain containing this water body is called "aquifer". Karst water aquifers represent the most important water reservoirs on our planet, furnishing a high percentage of the drinkable water feeding our aqueducts [90].

Karst processes are governed by the following simple chemical equation:

$$CaCO_3 + CO_2 + H_2O \leftrightarrow Ca(HCO_3)_2 \tag{2}$$

In this work, we describe some peculiar karst morphologies that originate when a karst system is fed not only by meteoric waters (autogenic karst), but also by runoff waters coming from a non-karst area (allogenic karst; [86]). This phenomenon is possible when limestone successions crop out in association with other less permeable or impermeable rocks. During meteoric events, the running waters collect on impermeable rocks and create a drainage network that flows on the surface until it meets soluble rocks. At this point, water may attack the more fractured rocks by chemical dissolution where it starts to create a concentrate infiltration point, the so-called ponor. Thus, what are ponors? They are simply holes on the Earth's surface where streams disappear underground. They are also called swallow holes or stream-sinks. The stream waters, during geological time, are able to "dig" these holes in the carbonate rocks because they are soluble and susceptible to karst processes. This kind

of ponor, located at the contact between permeable and impermeable rocks, is referred to by geologists as a "contact ponor" (Figure 4; [70]). The largest is the upstream valley, and the biggest will be the hole that the water may dig in the carbonate rocks. The drainage basin located upstream of a ponor is called the "blind valley" because it has no continuation on the surface, but it disappears underground. In association with ponors there are always cave systems (Figure 5A) that, depending on the difference in altitude between the ponor and the basal water table, may transfer the surface water towards a resurgence or may directly feed the water table (Figure 5B). This condition makes karst aquifers highly vulnerable regarding possible contamination between surface waters and underground waters [91]. Anything spilled anywhere in the catchment of a blind valley may directly reach an underground water table, even if it is very far from the spilling point.

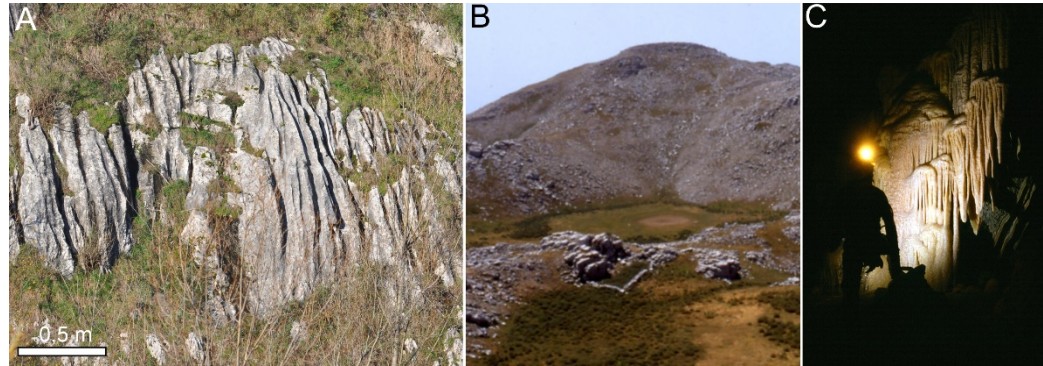

**Figure 3.** Examples of epikarst and endokarst morphologies in the Cilento, Vallo di Diano and Alburni Geopark: (**A**) karren; (**B**) doline at the summit of the Mt. Cervati peak; (**C**) carbonate concretion, stalactites, in a cave.

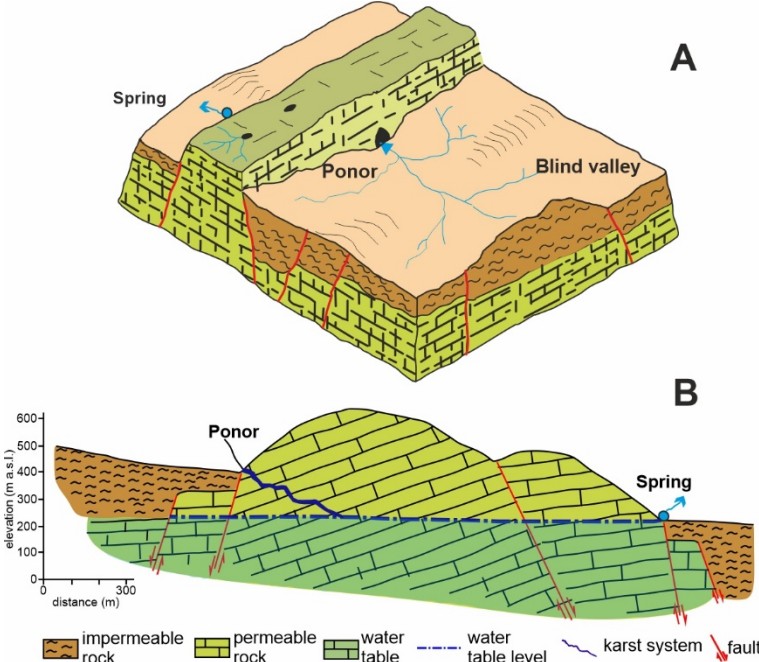

**Figure 4.** A 3D sketch (**A**) showing the formation of contact ponor and (**B**) cross-section view showing the communication between the stream waters sinking in the ponor and the basal water table.

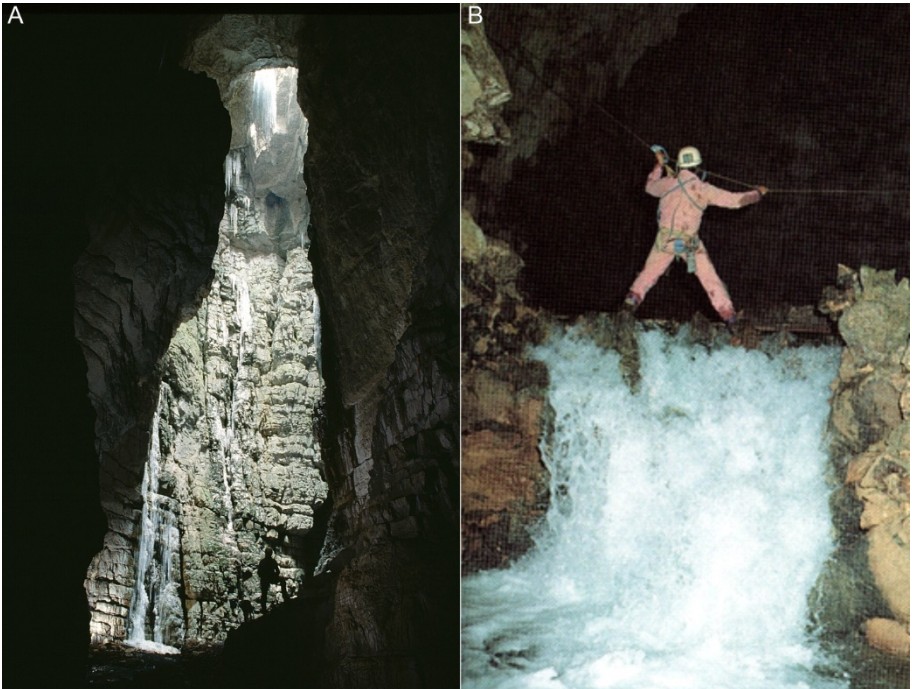

**Figure 5.** (**A**,**B**) Internal views of karst systems associated with ponors (Cilento, Vallo di Diano and Alburni Geopark).

In the southern Apennines, the main springs that are intercepted for drinkable use are placed at the base of the carbonate massifs, with single discharge that often exceeds 2000–3000 L/s [68,92]. Most of these water resources are fed by the karst system for which a communication between ponor at the surface and subsurface water table has been proven [93]. Amazing examples of these features are the endoreic karst basin of the Matese Massif and the Picentini Mts. [64,93,94] and the carbonate slopes of the Cilento, Vallo di Diano and Alburni Geopark [70,95]. In some cases, the contamination of the water sinking in the ponor has been proven [95,96]. This is what makes karst areas, where stream sinks and blind valleys are present, extremely sensitive environments, where surface waters may come directly in contact with underground water reservoirs. All human activities insisting on a drainage basin located upstream of a ponor must take into account the possibility of interfering with the basal water table. These simple concepts on how a karst aquifer functions should be basic components in the environmental education of every citizen and administrator. Geotourism offers the possibility of coming directly in contact with these problems, not only by studying but also in such activities as taking a walk. Therefore, we propose a geoitinerary in the area of the middle Bussento River system since it is a good case to understand this karst environment.

*4.2. The Karst Ponors and the Blind Valley of the Bussento River: Proposed Geoitinerary*

In the middle Bussento river valley karst processes are well developed and mostly represented by contact ponors with associated cave systems and blind valleys. The Bussento River originated from Mt. Cervati and Mt. Figarola (Figures 1 and 2). Its drainage basin is carved on impermeable and soft rocks belonging to Miocene flysch deposits (Cilento group) and to Eocene-Miocene clayey basinal successions (internal units). The river flows for about 10 km in a N-S direction on this soft rock then it reaches the northern slope of the carbonate ridge of Mt. Pannello and abruptly disappears, sinking into a big hole called the La Rupe ponor (Figure 6).

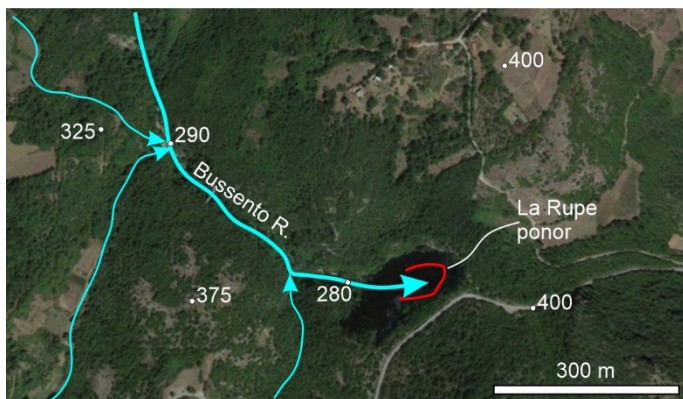

**Figure 6.** Google Earth image of the La Rupe ponor area. Arrows indicate flow direction. Red line indicates the La Rupe ponor. White points are elevation points.

The proposed geoitinerary is shown in Figure 7, whereas its technical issues are reported in Table 2.

**Table 2.** Technical issues of the proposed itinerary.

| The Middle Bussento Karst System Geoitinerary | | | | |
|---|---|---|---|---|
| Length | Duration | Number of Stops | How to Move between Stops | Type of Itinerary (According to Italian Excursionist Federation classification, [97]) |
| ~17 km | It could be completed in a one-day itinerary (expected duration is 8 h) or split in a two-day itinerary | 5 | by walking or by car | excursionist |
| **Stop no. 1: La Rupe ponor (3 km far from the urban center of Caselle in Pittari)** | | | | |
| **Path Length** | **Duration** | **Difference in elevation** | **Main geological and geomorphological features** | |
| 1 km | 2 h | 180 m | La Rupe ponor; Bussento River blind valley; dissolution pans; karren | |
| **Stop no. 2: Orsivacca ponor (0.8 km far from stop no. 1)** | | | | |
| **Path Length** | **Duration** | **Difference in elevation** | **Main geological and geomorphological features** | |
| 300 m | 1 h | 40 m | Orsivacca ponor; Orsivacca River blind valley | |
| **Stop no. 3: Rio della Bacuta valley (1.6 km far from stop no. 2)** | | | | |
| **Length** | **Duration** | **Difference in elevation** | **Main geological and geomorphological features** | |
| 500 m | 1 h | 60 m | Cozzetta ponor; Bacuta ponor; Rio della Bacuta River blind valley | |
| **Stop no. 4: MU.VI. (3.2 km far from stop no. 3)** | | | | |
| **Duration of the visit** | | **Main geological and geomorphological features** | | |
| 1 h | | Virtual tour of the Middle Bussento Karst System | | |
| **Stop no. 5: Bussento Resurgence (6.5 km far from stop no. 5)** | | | | |
| **Length** | **Duration** | **Difference in elevation** | **Main geological and geomorphological features** | |
| 700 m | 2 h | 120 m | Bussento Resurgence; Bussento gorge; fossils of rudists; Mulino spring | |

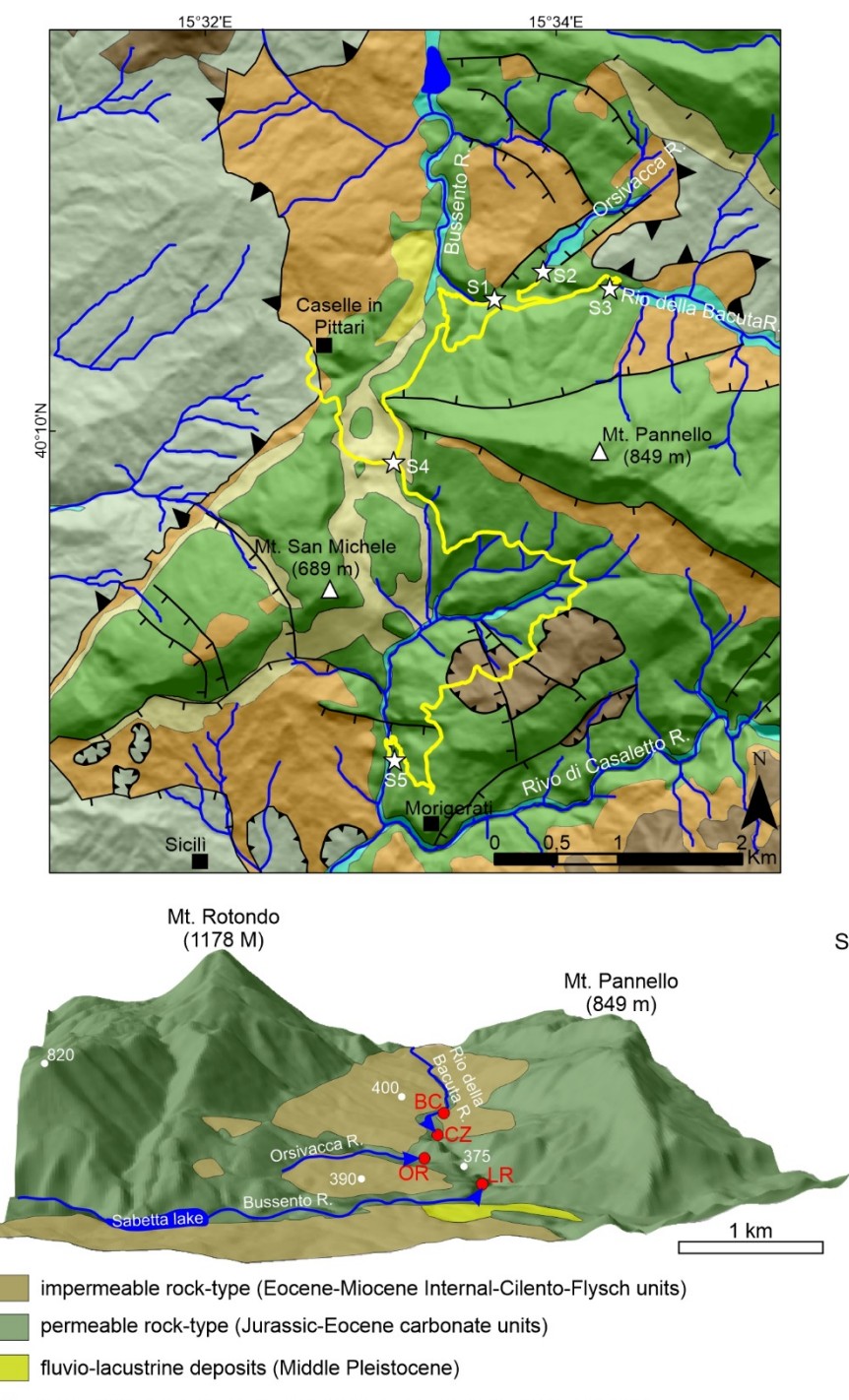

**Figure 7.** Upper panel: proposed geoitinerary in the discover of the Middle Bussento Karst System (yellow line). Stars indicate geoitinerary stops. S1: La Rupe ponor; S2: Orsivacca ponor; S3: Rio della Bacuta valley with its active (Bacuta) and fossil (Cozzetta) ponors; S4: MU. VI.; S5: Bussento Resurgence. See Figure 2 for legend of the geological units. Lower panel: 3D view, from west, of the area between the Bussento River, the Orsivacca River, and the Rio della Bacuta River. Red points are ponors. Blue lines indicate the river network and arrows indicate the flow direction. White points are elevation points.

The map and the 3D model of Figure 7 show that two little streams occur to the left of the Bussento River, named the Rio della Bacuta River and the Orsivacca River, which disappear underground and

may be classified as blind valleys. Their drainage basins upstream of the ponors are carved into soft rocks and the streams disappear when they meet the carbonate rocks.

The starting point of the path that bring tourists to stop no. 1, the La Rupe ponor (Figure 8A), is 3 km far from the urban area of Caselle in Pittari and it is near a gentle surface placed at the foot of the northern slope of Mt. Pannello. During the drop towards the La Rupe ponor, tourists may admire the stratified carbonate bedrock with several small-scale epikarst features (e.g., dissolution pans, karren). Tourists will reach the Bussento River valley bottom after having travelled for 900 m. The water level is very low because it is regulated by the Sabetta Lake hydroelectric powerplant placed about 2.5 km upstream of the Bussento ponor. Before the dam construction in 1958, the river waters entered the ponor with a higher flow rate, carrying out a significant debris load, as indicated by the large (cubic decimeters to cubic meters), rounded to sub-rounded boulders covering the valley floor. During extreme floods, this material obstructed the entrance of the ponor causing the formation of a wide lake. Local inhabitants called this phenomenon "Votamare" or "Ultimare", which means the area looked like the sea. After a 100 m walk along the Bussento River valley floor, tourists will reach the La Rupe ponor. The entrance, which is about 30 m high and 10 m large, is very spectacular and represents the point where the Bussento river starts to flow underground (Figure 8B,C). During geological time the river waters created an underground channel, at least 4 km long, that cut across the Mt. Pannello ridge and resurged at the surface at the foot of its southern slope, near the village of Morigerati, in the so-called Bussento Resurgence (Figure 7). Only tourists with either a speleological background or under the guide of an expert speleologist may visit the initial part of this underground channel, which is explored for only 566 m. It develops mainly following SW-NE and NW-SE trends and ends in a siphon lake.

From the La Rupe ponor, tourists will return to the main road to reach the Orsivacca, Cozzetta, and Bacuta ponors (stops S2 and S3 in Figure 7). The Orsivacca valley floor is dry for most of the year but during autumn and winter seasons it is possible to admire the water falling within the ponor. As shown in Figure 9, the ponor's dimensions are very little in respect to those of "La Rupe", being smaller their catchments. These ponors are in connection with caves and, in these cases, only persons with a good speleological background and under the guide of an expert speleologist have access.

After visiting the Rio della Bacuta and Orsivacca blind valleys, tourists will return to the main road to reach the fourth stop of the geoitinerary, the MU.VI. (Museo Virtuale, "Virtual Museum", stop S4 in Figure 7). The MU.VI. is a building (Figure 10) where it is possible to admire a permanent virtual exhibition about the Middle Bussento Karst System under the guide of experts either from local associations addressed to the promotion of the territory or from speleologists of the Italian Alpine Club (a speleological group). The visit to the MU.VI. is 1 h long and here it is possible to organize other outdoor activities (e.g., trekking, canyoning, pedal cars, mountain biking) that are not included in the proposed itinerary, but that can be provided by local associations. In addition, for those that enjoy outdoor activities, there is an area where it is possible to practice fitness exercises right in front of the MU.VI. entrance.

Tourists will then move from the MU.VI. towards the last stop of the geoitinerary, the Bussento Resurgence (stop S5 in Figure 7), which is 6.5 km away and can be reached by car (15 min) or by walking (~1 h).

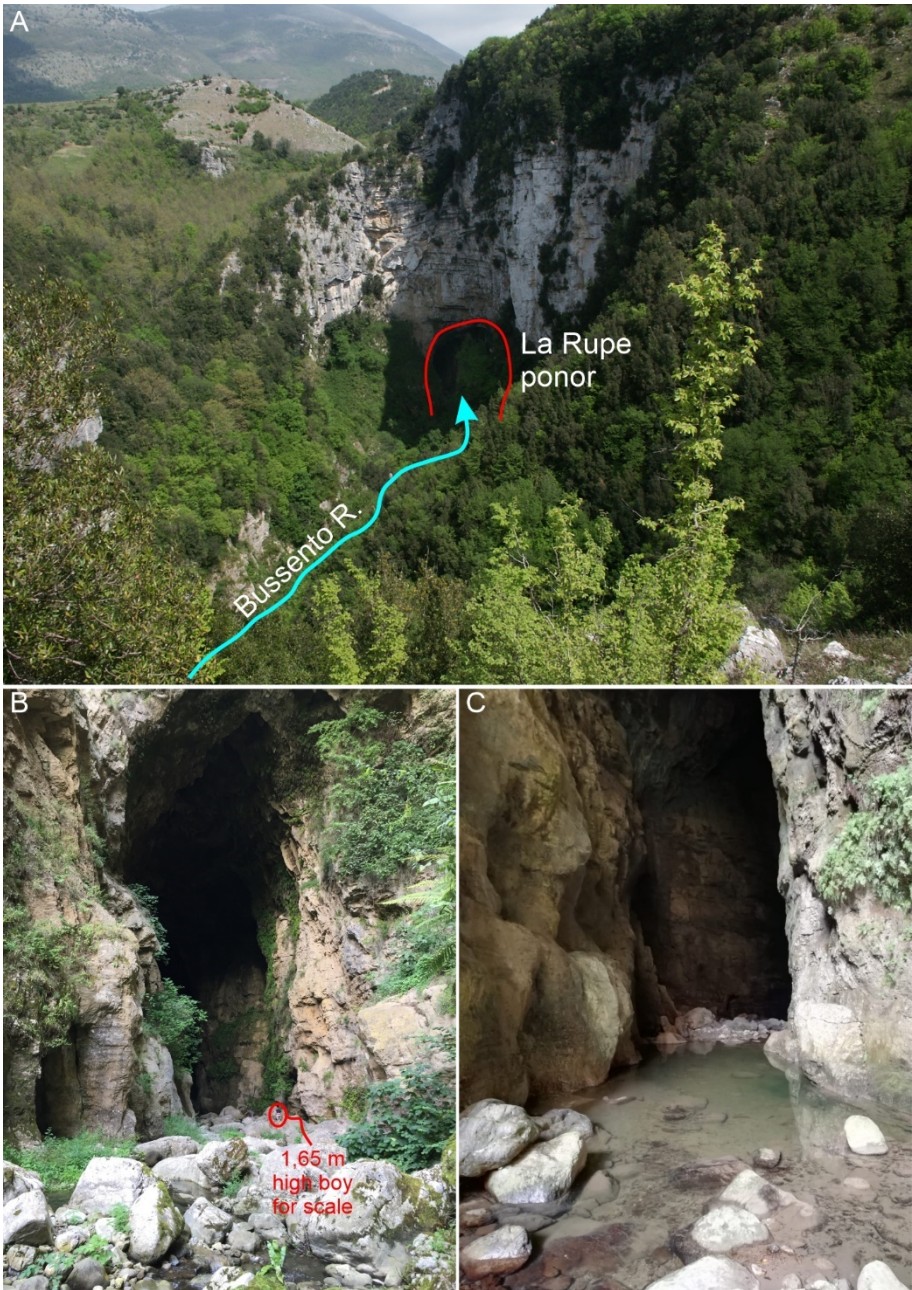

**Figure 8.** Stop no. 1: La Rupe ponor. (**A**) Panoramic view of the Bussento River valley and the entrance of the La Rupe ponor; (**B**) the La Rupe ponor entrance; (**C**) close view of the La Rupe ponor entrance.

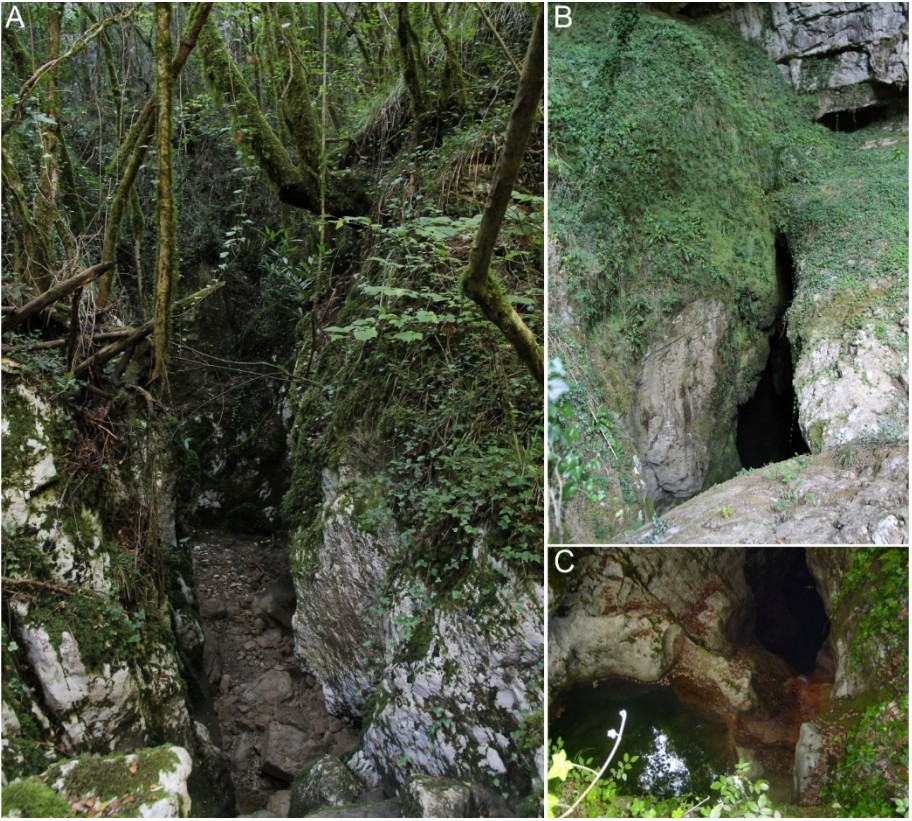

**Figure 9.** (**A**) Stop no. 2: the Orsivacca ponor; (**B**) the fossil Cozzetta ponor; (**C**) the active Bacuta ponor.

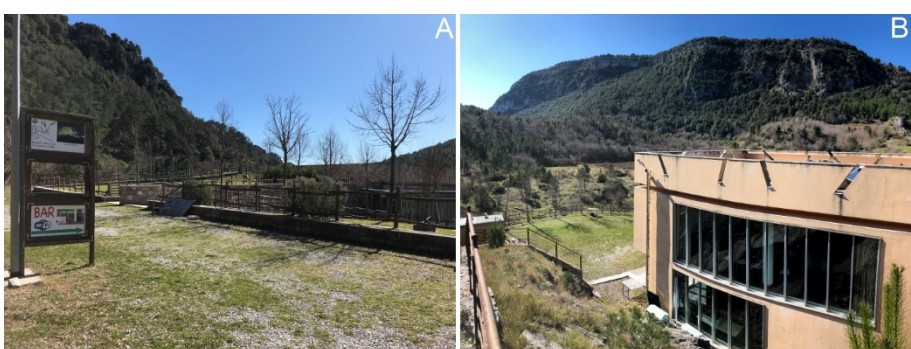

**Figure 10.** (**A**) Stop no. 4: the entrance of the MU. VI. (virtual museum); (**B**) lateral view of the MU.VI. with the eastern slope of Mt. San Michele in the background.

This stop visits both the Old Mill springs and the Bussento River resurgence. Along the drop towards the valley floor, tourists can admire amazing rudists fossils (Figure 11A) in the carbonate bedrock. A small railway is also present to facilitate the tour either for tourists not used to trekking or for people that do not want to become tired. Once tourists reach the valley floor, they may admire the amazing Old Mill spring (Figure 11B) which is one of the basal springs of the Salice-Coccovello carbonate aquifer. This spring has a mean discharge of 50 L/s. Then, walking along a woody path next to the water level, they may reach the Bussento Resurgence, the point where the Bussento River rises again to the surface. The cave system associated with the resurgence (Figure 11C) is mostly horizontal and expert speleologists explore it for 462 m. Hydrogeological analysis (tracing tests) has proved a subsurface connection between the La Rupe, Orsivacca, Cozzetta, and Bacuta ponors with the Bussento Resurgence, as explained in Figure 12. The waters sinking in these ponors connect and travel underground at least for 4 km, making this case the only example of an underground river in

Southern Italy. Considering the ponors and the resurgence altitude in respect of the water table level, it is possible to understand that there is the possibility that the stream waters during their underground path may feed the basal water table (see blue arrows in Figure 12).

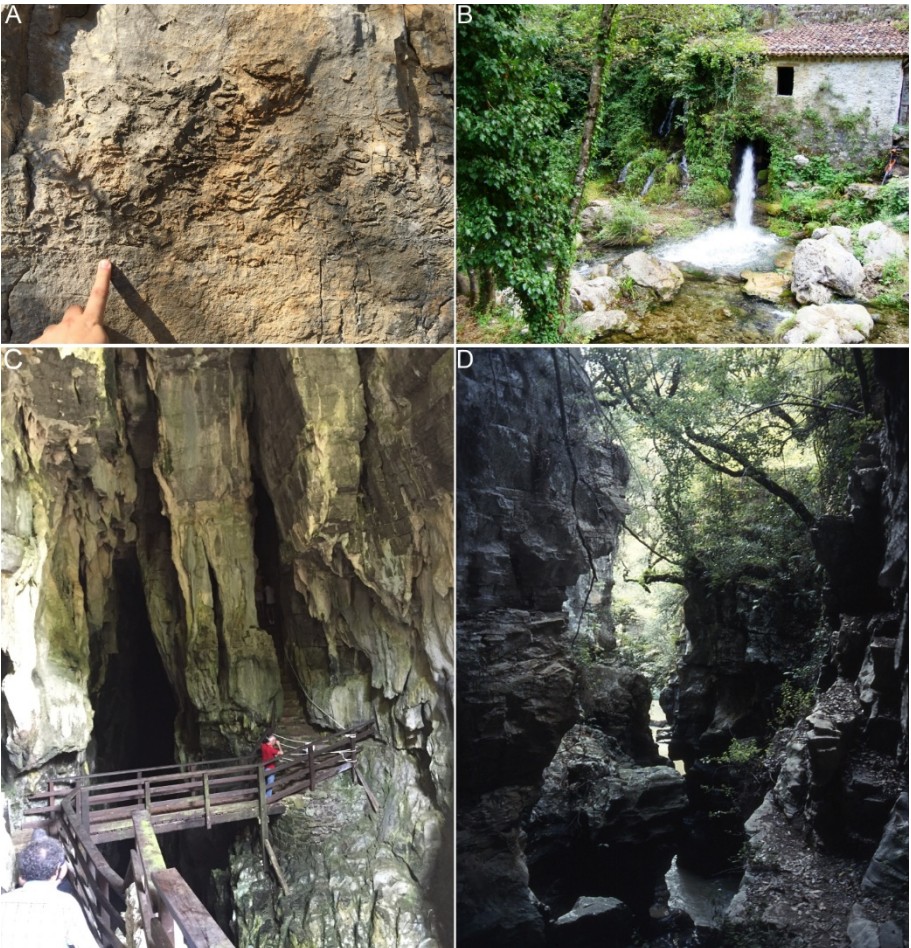

**Figure 11.** Stop no. 5: the Bussento Risurgence near Morigerati. (**A**) Detail of rudists fossils in the carbonatic bedrock; (**B**) the Old Mill spring; (**C**) the visiting tour in the interior of the Morigerati Resurgence; (**D**) view, from the inner side of the Bussento Resurgence, of the steep valley flanks.

The Bussento Resurgence, in addition to being an important geomorphosite, it is also a World Wildlife Foundation (WWF) Oasis, with an educational and international center, where it is possible to gain information about both the abiotic and biotic components of the environment.

Moreover, from the resurgence down-valley, the Bussento River created a gorge (Figure 11D), another peculiar karst morphology. Gorges are formed when a river flows on karst carbonate rocks and tends to dissolve them, leading to the formation of deep and narrow valleys, which often host a natural habitat of high environmental quality. Gorge flanks are high and very steep, making the gorges inaccessible areas except for expert speleologists or for those practicing canyoning.

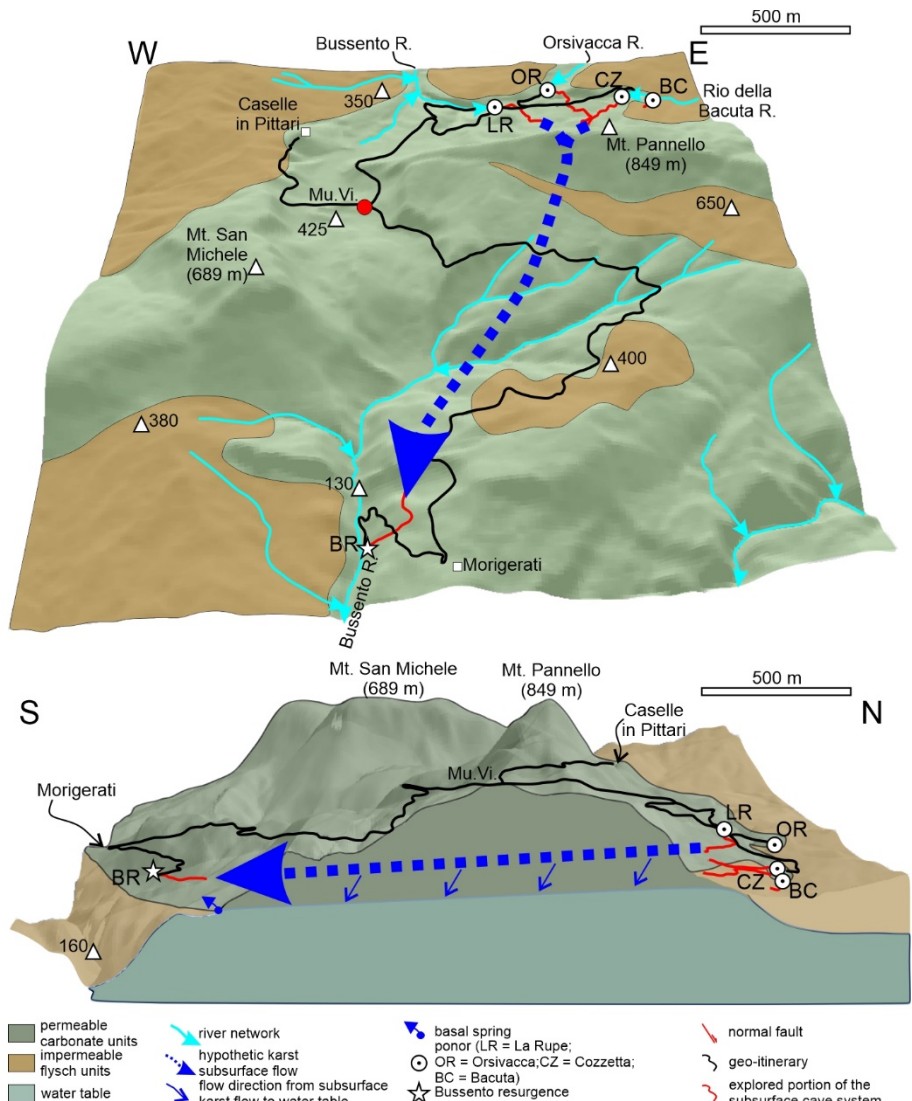

**Figure 12.** Frontal (from south, upper panel) and lateral (from east, lower panel) 3D views of the Middle Bussento Karst System. Red lines indicate the explored subsurface portion of the karst system (from [98]). White triangles in the upper panel are elevation points (expressed in meters above sea level).

## 5. Discussion

### 5.1. Geotourism in Karst Area

Karst terrains are widespread all over the world and cover about 17% of the Earth's surface [99], a percentage that increased up to 21.6% in Europe [100]. These data highlight the importance of karst terrains and landforms as crucial geological and geomorphological features to promote geotourism; in particular considering that 37% of Geoparks in the world exhibit karst phenomenon [101]. Ruban ([101] and references therein) highlighted the main features that make karst area prone to geotourism. According to Ruban, karst areas are, in fact, either unusual and rare landforms or windows into the dynamic geological environment: they attracted tourists before the actual concept of geotourism developed; they have aesthetic attractiveness; they are strongly related to the archaeological, historical, and ethnocultural peculiarities of some areas; they are of socio-economic importance because they are the main water reservoirs. Among karst landforms, the ones that play a major role in attracting tourists are caves [102]. Kim et al. [103] highlighted that cave tourism within geotourism gained large

popularity in Korea. Accordingly, Calaforra and Cortes [99] strengthened the importance of karst caves in geotourism as an instrument to favor local economic growth, indicating that karst caves in Spain receive more than 2.5 million visitors per year, with an economic return for the related municipalities (which often have less than 5000 inhabitants) in excess of 15 million euros [104]. Doorne [105] also highlighted the importance of karst caves in the local economy by discussing the Waitomo Glowworm Cave in New Zealand which serves a village of 500 inhabitants with a tourist population of around 450,000 international visitors per year. Allan et al. [106] focused on the motivations that lead tourists to visit places of geological importance by providing a questionnaire to tourists that visited the Cristal Cave in Australia. They found that the motivations of the visit include relaxation, escape from daily routine, sense of wonder, and knowledge. Hurtado et al. [107] also investigated Cristal Cave tourist traffic. Kiernar [108] discussed nature protection and geotourism in Laos where several karst caves receive tourist traffic related to religious motivation.

It is important that tourist traffic in karst caves avoid alteration of the natural environment, as proven for the Cave of Marvels in Spain [109]. Moreover, Baker and Genty [110] highlighted that tourist traffic in karst caves where ventilation is poor may alter $CO_2$ concentrations and increase temperatures by 3°. Calaforra et al. [111] carried out experimental measurements of caves' temperature at the Cueva del Agua de Iznalloz, Granada, Spain before the cave was opened to tourists, to determine the impact of human presence within the cave. Eagles et al. [112] pointed out that good planning and appropriate practices can lead to sustainable management of tourism.

Williams [113] stressed the importance of preserving the integrity of karst areas as a mandatory task because karst systems are complex systems that develop both at the surface and subsurface. Moreover, karst environments are an example of a fragile ecosystem whose balance depends on several factors such as the energy and quality of water flow. Williams [113] also remarked that environmental conditions in the recharge areas, both in allogenic and autogenic karst systems, have a strong influence on environmental conditions in the subsurface, thus suggesting that correct management of these areas is mandatory to avoid pollution in karst caves with dramatic consequences for plant and animals living in there. The previous point is fundamental considering that surface and subsurface water divides do not necessarily correspond, thus, karst drainage areas are not easy to delimit. The problem of preserving the integrity of a karst area is relevant also in the Middle Bussento Karst System. In fact, the Bussento River suffers both from low discharge due to the activity of the Sabetta Lake hydroelectrical powerplant and the organic pollution. This could cause severe problem to the 43 species found during speleological exploration at the La Rupe ponor [114,115]. Fortunately, subsurface water auto-depuration, due to a still poorly known subsurface karst path, dilutes the organic contaminants, making the water flowing out at the Bussento Resurgence in Morigerati the cleanest and able to host many fluvial organisms [114].

For what we have discussed up to now, it is evident that for geotourism in karst areas there is a fundamental task to diffuse the concept of environmental protection in these fragile ecosystems. The geoitinerary we propose intends to make people aware about the importance of karst areas as the main source for drinkable water by the combination of outdoor (trekking), indoor (speleological exploration under the guide of expert speleologists), and educational (visit to the MU.VI.) activities. It should also serve as a promoter of the territory by increasing curiosity about this poorly known portion of the Geopark and thus contributing to local economic growth.

*5.2. Promoting Geotourism in the Middle Bussento Karst System: What Has Been Done and What Can Be Done*

The scientific and educational values of the Middle Bussento Karst System are well known since the first exploration of the La Rupe ponor in the early 1950s [116]. Explorations were carried out in a discontinuous way until 2007 when speleologists from the Italian Alpine Club, section of Naples, reached the farthest point of the subsurface path [116]. Due to the high scientific value of the area, the Speleological Group of the Italian Alpine Club, section of Naples, in combination with other speleological groups from the Campania Region, organized the 2nd Regional Conference on Speleology in Caselle in Pittari, from 3–6 June 2010 [117]. This conference has been, up to now, the most important

scientific event during which the Middle Bussento Karst System has been at the center of discussions of hundreds of researchers and persons interested in speleology coming from all over Italy. During this conference, an intense educational activity was carried out at the MU.VI. by speleologists of the Campania Speleological Federation and the Speleological Group of the Italian Alpine Club, section of Naples [118].

Since then, educational activities geared towards knowledge and promotion of the territory have been carried out in a discontinuous way by local administrations and associations. These activities include outdoor sports such as canyoning, trekking, mountain biking, and pedal cars mainly concentrated during the spring and summer. In addition, local associations carry out environmental education activities in elementary, middle, and high schools in the surroundings of Caselle in Pittari. This often includes a visit of scholars to the MU.VI., where a permanent virtual exhibition on karst is present the entire year.

Further activities addressed to discover the Middle Bussento Karst System include personal initiatives by environmental guides often coming from areas outside the Geopark. These activities include a one-day trek to visit either the La Rupe ponor or the Bussento Resurgence. They usually do not include a one-night stay in the area to enjoy the hospitality of the locals and the amazing local food; thus, they do not contribute significantly in the growth of the local economy.

In our opinion, what is done to promote the territory, help the growth of the local economy, and emphasize the unicity of the Middle Bussento Karst System is good, but it is not enough. We must always have in mind that the Bussento River is the second longest subsurface path in Italy and this point should be a crucial promoting element to bring visitors to this area. The sporadic initiatives carried out either by local administrations and associations or environmental guides enhance the lack of coordination between these groups that, together, could do a lot for the local community. There is also a scarce amount of information on the Internet. For example, the local administration created a website to promote the area [119] but the amount of data available on the Internet about the Middle Bussento Karst System is sharply lower compared to other karst systems in the same Geopark (e.g., the Pertosa and Castelcivita caves).

We think that more convincing advertising actions are necessary if local administrations really want to help in the growth of local economy by attracting more tourists. The proposed geoitinerary emphasizes the fascinating subsurface world that characterizes the area, and tries to increase awareness among tourists, local people, and local administrations about the importance of karst areas as resources of potable water. In addition, the geoitinerary must be accompanied by a more effective and pervasive presence on the Internet through social media, and the production of attractive educational material that could make the Middle Bussento Karst System easily understood by non-geologists. An example could be the 3D reconstruction in Figure 12 that aims to describe the karst system in a simple way, helping tourists figure out the connection between the La Rupe, Orsivacca, Cozzetta, and Bacuta ponors and the Bussento Resurgence.

### 5.3. SWOT Analysis and VSG Index

To analyze the potential of the area and to highlight possible activities addressed to the efficient and effective use of the proposed geoitinerary, we carried out an analysis to define the strengths, weaknesses, opportunities, and threats (SWOT analysis). Results of the SWOT analysis are reported in Table 3.

**Table 3.** Results of the SWOT (strengths, weaknesses, opportunities and threats) analysis.

| Strengths | Weaknesses |
|---|---|
| 1. Discovery of a poorly known portion of the Cilento, Vallo di Diano and Alburni United Nations Educational, Scientific and Cultural Organization (UNESCO) Global Geopark.<br>2. The area is easily reachable by public transport.<br>3. The Bussento River is the second longest subsurface river in Italy and the longest one in Southern Italy.<br>4. High didactic value to explain the relation between surface and subsurface flows in karst areas.<br>5. Karst landforms are visible and comprehensible to tourists, even if they have no geological background. | 1. The geoitinerary is at an initial stage and deserves the development of further activities, such as explanatory panels for each stop.<br>2. The geoitinerary is a personal initiative by the Authors and lacks, up to now, enough of an intense collaboration with local administration.<br>3. The full development of the geoitinerary needs funding and a management policy.<br>4. Local people and local administration have not yet fully understood the high touristic and didactic potential of the area.<br>5. Tourists reach the study area because of the amazing local food but they have no cognition of the karst system in the surroundings.<br>6. Local accommodation facilities can host very little tourist traffic. |
| **Opportunities** | **Threats** |
| 1. The geoitinerary could be integrated in a wider tour because of its closeness with the main touristic attraction of the Geopark.<br>2. The geoitinerary could lead many tourists to move from the touristic coastal areas of Cilento to the poorly known inner areas of the Geopark.<br>3. This tourist flow could help the growth of the local economy (e.g., by increasing the number of accommodation facilities).<br>4. The growth of the local economy could provide work for young people and so it could reduce the abandonment of small mountain villages.<br>5. The geoitinerary could be split in two days thus providing enough time for the tourists to enjoy the hospitality of local people and food of excellent quality. | 1. The high and steep carbonate walls near the La Rupe ponor may be affected by rock falls and deserves mitigation actions.<br>2. Some mule track and wooden paths need maintenance.<br>3. Visits to the La Rupe ponor is strongly influenced by the activities of the Sabetta Lake hydroelectrical powerplant and is not allowed during basin emptying.<br>4. The Bussento River at the La Rupe ponor suffers both from the low discharge due to the activity of the Sabetta Lake hydroelectrical powerplant and the organic pollution.<br>5. This could cause severe problems for the 43 species discovered during speleological exploration at the La Rupe ponor. |

We then calculated the VSG (value of a site for geotourism) index by applying the Pica et al. [88] method reported in Table 1. Moreover, the Middle Bussento Karst System has a representativeness (RP) value of 5 because it is an amazing example of a subsurface river path with contact ponors representative of the ponor retreat mechanism, and its interest falls in many disciplines, including geomorphology, hydrology, and structural geology. The rareness (RR) index has a value of 5 because subsurface river paths are not so common in Italian territory and the Middle Bussento Karst System is the second longest subsurface river in Italy. The scenic-aesthetic (SCE) value is 5 because viewpoints are common along the entire geoitinerary and cromatic contrast is excellent, thus allowing a full appreciation of the karst landforms. The historical-archeological-cultural index (SAC) has a value of 3 because of restriction laws related to the protected area and to a poor connection with local tradition. The accessibility index (AC) has a value of 3 because the site is easily accessible by car and public transport, but it lacks services close to the stops of the geoitinerary.

The resulting value of the VSG index is 21 suggesting that the Middle Bussento Karst System has a high potential for geotourism.

## 6. Conclusions

Karst systems are sensitive environments that deserve accurate management and promotion strategies to avoid contamination of water resources used for drinkable needs and to avoid alteration

of the environment, with drastic consequences for the flora and fauna that live in these areas and also for the karst landforms as well. Furthermore, karst landforms are among the more fascinating landforms to attract tourists.

To highlight the natural and socio-economic role of karst areas, we investigated the Middle Bussento Karst System, in the Cilento, Vallo di Diano and Alburni UNESCO Global Geopark by carrying out a comprehensive analysis of both its surface and subsurface karst landforms. The peculiarity of the area is that the Bussento River sinks at the La Rupe ponor, near the village of Caselle in Pittari, and remerges, after 4 km of subsurface path, at the Bussento Resurgence, near the village of Morigerati, making the Bussento River subsurface path the second longest one in Italy and the longest one in Southern Italy. Selected karst landforms include the La Rupe, Orsivacca, Cozzetta, and Bacuta ponors, whose subsurface paths are connected and end at the Bussento Resurgence. In addition, a visit to a local virtual museum (MU.VI.) is included in the geoitinerary. The SWOT analysis also enhances the touristic potential of the geoitinerary, highlighting the high potential of this area as a possible touristic attraction in the Cilento, Vallo di Diano and Alburni UNESCO Global Geopark. This is also testified by the high value of the VSG index, whose score of 21 places the geomorphosite among the areas with high potential for geotourism. Some touristic promotion activity has already been carried out, such as the foundation of the MU.VI., however more actions are necessary to strengthen the role of the Middle Bussento Karst System as a significant geo-touristic attraction. The proposed geoitinerary has high scientific and educational values that may help in the field of environmental education, making people aware about the importance of karst areas as containers of water resources. The combination of the proposed geoitinerary with educational materials could help visitors gain detailed knowledge about what a karst environment is and how it works. Future campaigns of touristic promotion in the Geopark should emphasize the role of the blind valley of the Bussento River as the second longest subsurface river path of Italy and the longest in Southern Italy. Moreover, the addition of possible outdoor activities carried out by local associations could help the growth of tourism.

We tried to contribute to an increase in the curiosity of this site, in order to promote this fascinating portion of the Geopark as a touristic attraction and aid in the growth of the local economy.

**Author Contributions:** Conceptualization, E.V. and N.S.; methodology, E.V. and N.S.; software, E.V.; validation, A.S., D.G., and N.S.; formal analysis, D.G.; investigation, E.V.; data curation, E.V.; writing—original draft preparation, E.V., A.S., and N.S.; writing—review and editing, E.V., A.S., and N.S.; visualization, D.G.; supervision, A.S. and N.S. All authors have read and agreed to the published version of the manuscript.

**Funding:** The research was funded by department research (DISTAR), Nicoletta Santangelo.

**Acknowledgments:** We wish to thank two anonymous reviewers whose suggestions helped to increase the quality of the paper.

**Conflicts of Interest:** The authors declare no conflicts of interest.

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
