# Peer review of "Geotourism in the Cilento, Vallo di Diano and Alburni UNESCO Global Geopark (Southern Italy): The Middle Bussento Karst System"

_resources, doi:10.3390/resources9050052_

Round 1

Reviewer 1 Report

Title

Please insert in the title the country. You remember that resources is an international journal.

Abstract

In the abstract I Don´t see the information about the methodology that you employed for you created the geoitinerary.

Keywords:

In the abstract I you indicated the study area has a lot of geomorphological features, but in the keywords you writed geosites but not geomorphosites. The question is: The geomorphological characteristics of geopark are geosites or geomorphosites? I  recommend the international papers about de diference between geosites or geomorphosites.

1-Introduction

Your paper is about the geotourism, but in the introduction you the references about what is geotourism or the different approximations (geology or geography) of the geotourism concept are insuficients.

Lines 52 to 55 you shows the geopark definition, but I recommend you read the characteristics of geoparks in the information of global geoparks network.

Line 79 to 70 I believe understand that you said the geomorphosites are geosites? In line 83 you repet thata 32% of geosites are geomorphosites. I´m not sure about this aspects are true. In the scientif literature about the geomorphosites the authors diference clearly between geosites of geomorphosites. 

Line 105 you say the area receive thousand of visitors. Have an official statistics?

line 138 there is an error in the word landforms

2-The Middle Bussento Karst System

ok, but I put Study area

3-Material and Methods

How many field work was necessary?

The field information that you collected, Which system employed?

Lines 218 and 219 you write 3d, but in the other part of the paper you write 3D. Please write only in the 3D form

4-Results

Karst landforms (ponors, blind valleys, resurgences and karst springs): importance for 223 environmental education

Lines 133 and 134, when you say the karst have the most important water reservoir of on our planet, the affirmation is your or no. If the affirmation is the others authors, please indicated the reference.

Figure 3. The quality of the images in no good, but I think that the resources  journal send me a pdf copy with low quality.

Line 273 to 274. I don't understand the possibility that offers the geotourism with the problemas that you describe in the paragraphs only when the visitors walk in the geopark.

Figure 4. The sketch are your property? Please you must be indicated.

Figure 5, the same problem of the figura 3

In general, I think that is preferable if you write about the karst processes and forms with special references to the study area.

The Karst ponors and the blind valley of Bussento river: proposed geoitinerary

Line 286 have an error the word Figg

Figure 7, please you must indicated the property

Line 307 please write 3D not 3d

Table 1 you indicate the length of the geoitineray is 17 km, but the visitor have the possibility walk only 2.5 km  length?

In the table 1 you write about the main geolgical and geomorphological features. But your paper is about of the geotourism. Do you think that the cultural heritage of the geoitinerary are  mencionated in the table?

Figura 12, please you must indicated the property

In general in each stops the natural heritage are very good represented. But I believe that you must put more attention to cultural heritage in each stop

5-Results

For international scientific paper is necessary the a part dedicated to the a discussion. I propose you the the name of this part is results and discussion.

In my opinion, the results has must expresed the relationship of the type of geoitinerary with others geoturistical itineraries in karsts or in others geological context.

What type of approximation of geotourism that you use in the paper?. Please read the Dowling R. & Newsome, D. (2018). Geotourism: definition, characteristics and international perspectives. En: R. Dowling R. & D. Newsome (Eds.), Handbook of Geotourism (pp. 1-22). Cheltenham: Edward Elgar.

What relationship have your geoitinerary with others? For example, in others geoitineraries the authors employed the same methodology for the selección the stops? The information in the stops are the same that other geotrails?

The SWOT analysis is result of your work? If  you say yes, the SWOT is correct in the discussión. But if you say not, the SWOT analysis is more apopiate for the introduction.

6-Conclusions

I believe that the conclusions is very poor. The conclusion not expresses the main aspects of the paper. In other hand, I thinks is necessary to add information about the future investigations in the same aspects: geoitineraries whit the special attention in the geotourism, the methodologiies employe....

References

The references are sufficient and actualised.

Reviewer 2 Report

Dear Authors,

I read the manuscript with interest. However, in my opinion its descriptive character is its serious weakness. No research methods have been used in it - for example, surveys or assessment of geosites. 

Therefore, the manuscript does not contain research results, discussion or conclusions. Even SWOT analysis is descriptive rather than tabular as usual.

It is not known why these 4 objects were chosen. Whether they are typical or unique to this area?

Therefore, it is difficult to consider the manuscript as a research paper. It is also not a review paper. 

I would suggest the following changes and additions:
- assessment of geotouristic values of geo-sites, included in the proposed route and located nearby, could be the basis for their selection;
- adding a broader (international) context to the manuscript - comparison of karst heritage in geoparks around the world; an indication of good practices in the field of sharing karst heritage;
- preparation of a synthetic table containing the results of the SWOT analysis;
- I would suggest reducing the size of photographs and maps in the manuscript

Round 2

Reviewer 1 Report

Dear authors, many thanks because you incorporated all of my suggestions or comments. In my humble opinion I believe that your work now is better.

Reviewer 2 Report

Dear Authors,

Thank you for making corrections to the manuscript. Most of my comments have been taken into account. As I wrote in the first review, I find the manuscript interesting although I would expect a more extensive research aspect. And not just the description.

However, if the assessment of geotouristic values of studied geosites was performed, its results should be included in the text. You should refer ratings of your sites to "average" ratings for other geosites within the geopark. This would confirm the weaknesses and strengths of the karst heritage.
